# Anatomic Origin of Osteochondrogenic Progenitors Impacts Sensitivity to EWS-FLI1-Induced Transformation

**DOI:** 10.3390/cancers11030313

**Published:** 2019-03-06

**Authors:** Elise R. Pfaltzgraff, April Apfelbaum, Andrew P. Kassa, Jane Y. Song, Wei Jiang, Tahra K. Suhan, Deneen M. Wellik, Elizabeth R. Lawlor

**Affiliations:** 1Department of Pediatrics, University of Michigan, Ann Arbor, MI 48109, USA; erpfaltz@umich.edu (E.R.P.); aapfelba@umich.edu (A.A.); apkassa@umich.edu (A.P.K.); weijian@umich.edu (W.J.); tahraklu@umich.edu (T.K.S.); 2Cellular and Molecular Biology Program, University of Michigan, Ann Arbor, MI 48109, USA; janeys@umich.edu; 3Department of Internal Medicine, Division of Molecular Medicine and Genetics, University of Michigan, Ann Arbor, MI 48109, USA; 4Department of Pathology, University of Michigan, Ann Arbor, MI 48109, USA

**Keywords:** Ewing sarcoma, Hox, cell of origin, mesenchymal progenitor cells, MSCs

## Abstract

Ewing sarcomas predominantly arise in pelvic and stylopod bones (i.e., femur and humerus), likely as a consequence of *EWS-FLI1* oncogene-induced transformation of mesenchymal stem/progenitor cells (MSCs). MSCs located in the embryonic superficial zone cells (eSZ) of limbs express anatomically distinct posterior *Hox* genes. Significantly, high expression of posterior *HOXD* genes, especially *HOXD13*, is a hallmark of Ewing sarcoma. These data drove our hypothesis that *Hox* genes in posterior skeleton MSCs contribute to Ewing sarcoma tumorigenesis. We isolated eSZ cells from stylopod and zeugopod (i.e., tibia/fibula, radius/ulna) bones, from wild-type and *Hoxd13* mutant embryos, and tested the impact of *EWS-FLI1* transduction on cell proliferation, gene expression, and tumorigenicity. Our data demonstrate that both stylopod and zeugopod eSZ cells tolerate EWS-FLI1 but that stylopod eSZ cells are relatively more susceptible, demonstrating changes in proliferation and gene expression consistent with initiation of malignant transformation. Significantly, loss of *Hoxd13* had no impact, showing that it is dispensable for the initiation of *EWS-FLI1*-induced transformation in mouse MSCs. These findings show that MSCs from anatomically distinct sites are differentially susceptible to EWS-FLI1-induced transformation, supporting the premise that the dominant presentation of Ewing sarcoma in pelvic and stylopod bones is attributable to anatomically-defined differences in MSCs.

## 1. Introduction

Ewing sarcoma exhibits a characteristic pattern of primary tumor sites. The most common primary locations for Ewing sarcoma in adolescents and young adults are in the pelvis (26%) and the femur (20%) [1]. Very rarely are primary Ewing tumors found in the bones of the skull (2%) or hand (1%) [1]. We propose that this anatomic distribution can enhance our understanding about the enigmatic ontogeny of Ewing sarcoma, and hypothesize that cells of tumor origin are found more predominantly in the regions of the body more often afflicted with primary disease.

The early cellular and molecular origins of Ewing sarcoma are largely unknown. Tumorigenesis is driven by EWS-ETS fusion proteins, most often EWS-FLI1, and secondary genetic events are rare, revealing that, in the context of the human disease, EWS-FLI1 is both necessary and sufficient to induce malignant transformation [2,3,4,5]. Studies of in vitro and ex vivo transduction/transplantation experiments have shown that, while most primary cells do not tolerate EWS-FLI1 [6,7], select populations of mesenchymal stem and progenitor cells (MSCs) are permissive and capable of initiating transformation. Specifically, human bone marrow- and neural crest-derived MSCs tolerate EWS-FLI1 and induce an oncogenic transcription program, but are unable to form tumors when subcutaneously injected into immunodeficient mice [8,9,10,11]. In addition, despite diverse approaches, attempts to model EWS-FLI1-induced tumorigenesis in syngeneic transgenic mouse models have, to date, been unsuccessful [12]. Ex vivo transduction/transplantation models of isolated primary murine MSCs have been relatively more successful, but these models also demonstrate conflicting results, with evidence to both support and refute the claim that EWS-FLI1 can, by itself, transform primary MSCs [13,14,15,16]. Most recently, discrete osteochondrogenic progenitors isolated from embryonic superficial zones (eSZ) of E18.5 mice were shown to be highly susceptible for EWS-FLI1-induced malignant transformation [15]. Thus, cell context is of key importance and the factors that determine susceptibility of discrete bone-derived MSCs to EWS-FLI1-induced transformation remain to be elucidated.

We recently reported that high expression of posterior Homeobox D gene cluster (*HOXD)* genes is a hallmark of Ewing sarcoma and that ectopic expression of EWS-FLI1 in neural crest-derived MSCs hijacks normal epigenetic regulation of *HOXD10*, *HOXD11*, and *HOXD13* [17]. In addition, loss of posterior *HOXD* gene expression in established Ewing sarcoma cells leads to a reduction of tumorigenicity, confirming a role for these genes in tumor pathogenesis [17,18]. During embryonic development, axial patterning is based on the tightly regulated spatial and temporal expression of *Hox* genes. *Hox* genes are expressed in an anterior to posterior pattern with *Hox1* expressed in the most anterior aspect of the embryo and *Hox13* in the most posterior [19]. In the context of limb development, expression and function of *Hox* genes are restricted to spatial compartments so that the embryonic progenitors of the radius and ulna are (zeugopod) are *Hox11* positive, while those of the humerus (stylopod) are *Hox10* positive with the pelvis being both *Hox10* and *Hox11* positive [19,20]. Significantly, recent studies from our group showed that this discrete anatomic pattern of *Hox* gene expression extends throughout development and into adulthood and that these *Hox*-expressing cells are multipotent MSCs [19,21]. These data led us to test the hypothesis that the propensity for Ewing sarcoma to arise in the posterior skeleton is due to an inherent susceptibility of posterior *HOX* expressing MSCs to EWS-FLI1-induced malignant transformation.

## 2. Results

### 2.1. Regional Differences in Ewing sarcoma Presentation Mirror Regional Hox Compartmentalization

The most common young adult and adolescent presentation of Ewing sarcoma is in the bones of the pelvis and the femur. A side by side comparison of the frequency of localization of Ewing sarcoma to that of the pattern of developmental *Hox* gene expression reveals an interesting association (Figure 1A,B). Ewing sarcoma is largely restricted to distinct regions as defined by Hox expression, with an emphasis on posterior Hox genes. Among the proposed cells of origin for Ewing sarcoma are progenitors within the eSZ [15]. The eSZ is a layer of osteochondrogenic progenitor cells on the surface of bones undergoing endochondral ossification (Figure 1C). These progenitors give rise to the proliferative zone chondrocytes, which in turn become the hypertrophic zone chondrocytes and eventually contribute to trabecular bone. In the context of the EWS-FLI1 transformation model, the susceptible tumor-initiating progenitor cells in the eSZ were defined by their expression of Pthrp [15]. We recently showed that MSCs within the eSZ cells express high levels of regionally-restricted *Hox* genes [21]. To determine whether Pthrp+ eSZ progenitors are *Hox+* MSCs, we crossed *Hoxa11-EGFP* (green fluorescent protein) and *Pthrp*-LacZ reporter mice and examined developing eSZ regions by immunohistology. As shown, Pthrp+ eSZ cells in the zeugopod are *Hox11+* (Figure 1D,E), supporting the hypothesis that Hox+ progenitors in the developing bone may be differentially susceptible to malignant transformation by EWS-FLI1.

### 2.2. Posterior HoxD Genes Are Expressed by Isolated eSZ Cells

Having established that candidate Ewing sarcoma cells of origin, the eSZ progenitor cells, express Hox, we next sought to compare the impact of EWS-FLI1 on eSZ cells from different anatomic sites. The stylopod is one of the most common primary sites of Ewing sarcoma. The zeugopod is a skeletally similar, yet anatomically distinct region, in which Ewing sarcoma presents less frequently. We therefore evaluated whether stylopod-derived eSZ cells are more sensitive to EWS-FLI1 induced transformation compared to zeugopod cells.

eSZ cells were isolated from E18.5 stylopods (humerus and femur) and zeugopods (tibia/fibula and radius/ulna) and posterior *Hoxd* gene expression was measured in the two isolated cell populations by qPCR. We focused our studies on the *HoxD* cluster based on findings from our group and others which defined posterior *HOXD* genes, specifically, as being critical for Ewing sarcoma pathogenesis in humans [17,18]. Consistent with their developmental origins, the eSZ cells from the stylopod expressed higher levels of *Hoxd10* (Figure 2A), while cells from the zeugopod expressed higher levels of *Hoxd11* (Figure 2B). Zeugopod eSZ also expressed higher levels of *Hoxd13* (Figure 2C), likely reflecting the contribution of cells from the distal eSZ regions [22]. Consistent with immunohistochemistry, eSZ cells from both stylopod and zeugopod express Pthrp (Figure 2D).

### 2.3. EWS-FLI1 Transduction of Stylopod and Zeugopod eSZ Cells

Having established that *Hox+* eSZs could be reproducibly isolated from zeugopod and stylopod, we next sought to determine if one or both populations would tolerate lentiviral-mediated expression of EWS-FLI1. Cells were transduced with increasing concentrations of EWS-FLI1-EGFP lentivirus and, in keeping with prior reports [15], transduction efficiency of primary eSZ cells was approximately 20%. In addition, efficiency was statistically equivalent between empty vector control and *EWS-FLI1*-transduced cells, as well as between stylopod and zeugopod eSZs (Figure 3A). Two days post transduction eSZs were alive, were GFP+ by microscopy, and dose-dependent expression of EWS-FLI1 was confirmed by qPCR (Figure 3B). Expression of *EWS-FLI1* was confirmed, in independent experiments, to be equivalent between stylopod and zeugopod cells two days post transduction (Figure 3C). Thus, transduction efficiency and EWS-FLI1 expression levels were initially equivalent in stylopod and zeugopod eSZ, permitting comparison of oncogene-induced changes in *Hox* gene expression between these two populations.

After two days in culture in vitro, regionally-defined differences in Hox gene expression disappeared and expression of *Hoxd10*, *Hoxd11* and *Hoxd13* were no longer expressed in their regionally compartmentalized pattern (Figure 3D). In addition, this loss of differential expression was evident in both control and *EWS-FLI1* transduced cells, revealing that expression of *Hox* genes changes once eSZ cells are plated in culture, irrespective of the presence of the fusion oncogene. These data reveal the critical contribution of the microenvironment to maintenance of regional Hox expression patterns and suggest that EWS-FLI1 does not, at least in the short term, alter expression of posterior *HoxD* genes in murine eSZ cells.

Next, we measured the impact of EWS-FLI1 on expression of established target genes *Gli1*, *Prkcb*, and *Dkk2* that were previously reported to be induced by the fusion in the context of primary mouse eSZ [15]. Importantly, these genes are also targets of EWS-FLI1 in human Ewing sarcoma and contribute to tumor pathogenesis [23,24,25]. Notably, a trend to increased expression of both *Dkk2* and *Prkcb* was detected after two days in EWS-FLI1+ stylopod cells (Figure 3E). In contrast, no induction of any of these three target genes was observed in zeugopod eSZ cells.

### 2.4. Impact of Anatomic Origin on EWS-FLI1-Induced Phenotype

In human MSCs, initiation of the oncogenic transcription program and oncogenic phenotype by EWS-FLI1 requires days or even weeks [10,11,15]. Therefore, we maintained EWS-FLI1+ eSZ cells in culture for 12 days and evaluated changes in phenotype and gene expression. After 12 days in culture, EWS-FLI1+ stylopod cells displayed a growth advantage compared to empty vector controls and to both control and EWS-FLI+ zeugopod cells (Figure 4A). The growth advantage of the EWS-FLI expressing stylopod cells was not associated with any difference in the frequency of remaining GFP+ cells as this was found to be equivalent in all conditions (Figure 4B).

In parallel to the emergent growth advantage of EWS-FLI1+ stylopod cells, we noted a marked difference in cell morphology. Specifically, 10 days post-transduction, the EWS-FLI1+ stylopod cells were small and round compared to EWS-FLI1+ zeugopod and both control populations, which remained enlarged and more fibroblastic in appearance (Figure 4C). This change in morphology to smaller, more cuboidal cells is a characteristic feature of cells in which EWS-FLI1 has initiated the process of malignant transformation [8,10]. These differences in morphology between fusion+ stylopod and zeugopod cells were not associated with differences in expression of EWS-FLI1 (Figure 4D). By day 21, both control populations were large and flattened and they had adopted a senescent-like appearance. In contrast, EWS-FLI1+ stylopod cells continued to expand as small, round cells (Figure 4C). Notably, by 21 days the morphology of the EWS-FLI1+ zeugopod cells was equivalent to that of the EWS-FLI1+ stylopod cells suggesting that initiation of the EWS-FLI1-transformed phenotype is feasible in zeugopod eSZ cells but that the process takes longer and/or that the proportion of transformation-susceptible cells is lower in the zeugopod than in the stylopod.

We previously showed that EWS-FLI1 can induce upregulation of posterior HOXD genes in human neural crest-derived MSCs that have been passaged in culture [10,17]. In the primary eSZ cultures, we found that *Hoxd10* expression was increased in EWS-FLI1+ cells at day 12 compared with controls, and that up regulation was more statistically significant in zeugopod than stylopod cells (Figure 4E). Likewise, *Hoxd11* expression was highest in EWS-FLI1+ zeugopod cells (Figure 4F). *Hoxd13* was unaltered by EWS-FLI1, though expression remained higher in zeugopod cells and the lowest expression levels were detected in the EWS-FLI1+ stylopod cells (Figure 4G). Thus, stylopod cells are more susceptible than zeugopod cells to initiation of EWS-FLI1-induced transformation, as evidenced by a classic morphologic change and by demonstration of an early growth advantage. However, increased expression of posterior *HoxD* cluster genes was more often observed in EWS-FLI1+ zeugopod cells. These data suggest that EWS-FLI1-induced transformation of stylopod MSCs occurs independently of Hox gene activation.

### 2.5. Impact of Hoxd13 Loss on EWS-FLI1-Induced Transformation

Human Ewing sarcoma tumors over-express *HOXD10*, *HOXD11*, and *HOXD13* and maintenance of the tumorigenic Ewing sarcoma state requires continued high-level expression of *HOXD13* [17,18,26]. As shown above, eSZ cells express posterior *HoxD* genes and tolerate expression of EWS-FLI1. In addition, although these cells did not form tumors when injected into mice, other investigators have successfully demonstrated that stylopod-derived eSZ cells can be induced to full malignant transformation and that resulting tumors display genotypic and phenotypic properties of Ewing sarcoma [15]. Although its expression is confined to the autopod in post-natal development, *Hoxd13* serves as a master regulator of the entire posterior *HoxD* locus during embryonic limb development [27]. Thus, we sought to test whether expression of posterior *HoxD* genes and/or initiation of the EWS-FLI1-induced transformed phenotype in primary murine eSZ cells requires *Hoxd13*. To achieve this, we isolated E18.5 eSZ cells from stylopods and zeugopods of *Hoxd13* wild type (*Hoxd13^WT/WT^*), heterozygous (*Hoxd13^neo/WT^*), and mutant (*Hoxd13^neo/neo^*) embryos. Cells were transduced with EWS-FLI1 and changes in phenotype and gene expression monitored as described above. Knockout of one or both *Hoxd13* alleles was confirmed by genotyping and was evident in the dose-dependent expression of *Hoxd13* mRNA in wild-type, het, and null embryo-derived eSZ cells (Figure 5A). As with wild-type eSZ, we noted that EWS-FLI1+ stylopod cells had a growth advantage compared to EWS-FLI1+ zeugopod cells and this growth advantage did not segregate based on genotype. Specifically, after 12 days in culture, *Hoxd13* null eSZ cells were equivalent in number and in morphology to wild-type and heterozygous cells (Figure 5B). Moreover, there was no difference in transduction efficiency (Figure 5C) or *EWS-FLI1* expression levels (Figure 5D) among the three genotypes. The observed increase in *Hoxd10* and *Hoxd11* expression in EWS-FLI1+ zeugopod eSZ cells was again apparent and was irrespective of *Hoxd13* status (Figure 5E,F), demonstrating that *Hoxd13* is dispensible, both for cell tolerance of EWS-FLI1 and for activation of adjacent posterior *Hoxd* genes in both stylopod and zeugopod eSZ cells.

Having determined that the differential growth advantage of EWS-FLI1+ stylopod cells after 12 days in culture was neither dependent on *Hoxd13*, nor associated with enhanced expression of other posterior *HoxD* genes, we next evaluated expression of *Dkk2*, *Prkcb*, and *Gli1*, given that early changes in expression of these tumor-promoting genes had been apparent after two days (see Figure 3 above). Significantly, after 12 days in culture, EWS-FLI1+ stylopod-derived eSZ cells expressed higher levels of all three genes compared to EWS-FLI1+ zeugopod cells (Figure 5G–I). In particular, whereas *Prkcb* and *Gli1* levels had diminished to nearly undetectable levels in the zeugopod-derived cells, expression of both genes was significantly and reproducibly increased in EWS-FLI1+ stylopod eSZ cells (Figure 5H,I). These changes were not associated with *Hoxd13* genotype. These data suggest that differential activation of downstream oncogenic target genes, such as Prkcb and Gli1, may explain the differential susceptibility of stylopod MSCs to initiation of EWS-FLI1-induced malignant transformation.

## 3. Discussion

Ewing sarcoma remains an enigmatic tumor whose genetics have been well-characterized but for which a definitive cell of origin remains elusive. The inherent toxicity of the tumor-initiating EWS-FLI1 fusion oncoprotein has long challenged studies of primary cellular transformation and, to date, attempts to develop syngeneic mouse models have failed [6,7,8,12]. The cumulative evidence from multiple laboratories over the past 20 years supports the conclusion that Ewing sarcomas arise from early stem or progenitor cells of mesenchymal origin and that malignant transformation results as a consequence of EWS-FLI1-induced reprogramming of the epigenome [28,29,30,31]. Nevertheless, divergent results are observed between mouse and human cells, and when different MSCs are targeted, reveal the key importance of cellular context even within phenotypically defined progenitor populations [8,10,13,14,16]. Ewing sarcoma most often presents in the bones of the posterior skeleton, in particular, in the pelvic girdle and femur, suggesting that either EWS-FLI1 translocation events occur more commonly in MSCs in these anatomic sites and/or that regional differences in MSC biology impart differential susceptibility to transformation. Significantly, mammalian limb development is crucially dependent on the spatiotemporal functions of posterior Hox genes during embryogenesis, in particular *HoxA* and *HoxD* clusters, and the pattern and function of *Hox* genes is similarly regionally restricted in MSCs that are isolated from these anatomic sites [19,32]. The recent discovery that posterior HOXD genes are highly over-expressed by Ewing sarcomas [17], and function to maintain the tumorigenic phenotype [18,26], led us to hypothesize that cooperation between EWS-FLI1 and HOX genes plays an essential role in transformation of MSCs.

To begin to test this hypothesis we exploited the power of a recently described primary cell model of sarcomagenesis in which murine Pthrp+ progenitors isolated from stylopod eSZ were successfully transformed to Ewing-like tumors by EWS-FLI1 [15]. Significantly, crossing Pthrp-lacZ and Hoxa11-GFP reporter mice confirmed that Pthrp+ cells in the eSZ express high levels of Hox, lending initial support for our hypothesis that posterior Hox expression might contribute to malignant transformation. In keeping with the published model, stylopod-derived eSZ cells showed both phenotypic changes and a growth advantage upon EWS-FLI1-transduction, and, over time, induction of EWS-FLI1 oncogenic target genes was observed. Although these cells did not form tumors when injected into recipient nude mice, they did avoid senescence, unlike their control vector- transduced counterparts that failed to proliferate after several weeks in culture. In contrast to stylopod-derived eSZ cells, zeugopod cells were less susceptible to the transforming effects of EWS-FLI11. They showed no growth advantage and morphologic changes developed only after prolonged time in culture. In addition, our studies with *Hoxd13* mutant eSZ cells demonstrate that susceptibility to EWS-FLI1-induced transformation is not impacted by loss of *Hoxd13* revealing that, either the murine cellular context differs from human or, that *Hoxd13* is critical for tumor maintenance but dispensable for tumor initiation. Given that *Hoxa13* can compensate for loss of Hoxd13 function in development [32], it is possible that it may have served a similar role in these studies of malignant transformation. Studies with compound mutant eSZ cells will be necessary to definitively address this question. Likewise, given the unique susceptibility of the stylopod MCSs to EWS-FLI1, studies of *Hoxd10* function are also now highly worthy of pursuit. In the context of human Ewing sarcoma, knockdown of HOXD10, HOXD11, and HOXD13 all independently impaired the growth and invasive properties of tumor cells, demonstrating that despite the marked over-expression of *HOXD13*, all members of the posterior HOXD cluster contribute to human tumorigenesis [18]. Whether they work in concert or independently, and how and why they are coordinately activated in Ewing sarcoma, remains to be determined. Study of stylopod MSCs from *Hoxd10* mutant mice, alone and in combination with loss of function mutations in other posterior *HoxD* loci, will be useful to definitively establish the requirement of each of these genes to the initiation of the EWS-FLI1 transformed phenotype.

It is intriguing that, despite the given the marked enrichment of the Hox+ MSC in the epiphysis, the anatomic presentation of human tumors in stylopod bones is normally in the diaphysis. We hypothesize that there are two potential explanations for this discrepancy. First, although enriched in the embryonic superficial zone, Hox+ MSC are also present along the perichondrium of the diaphysis (see Figure 1D and [21]), and these cells would also be predicted to be susceptible to transformation. It is conceivable that the microenvironment of perichondrial MSCs is more conducive to cell transformation than that of epiphyseal MSCs, especially given proximity of diaphyseal MSCs to bone marrow. The bone marrow provides a rich and supportive tumor niche that favors Ewing sarcoma growth in human patients. A second possibility is that the cells of origin reside in the epiphyseal region at the time of the initial translocation event but that they migrate to the diaphysis prior to clinical presentation. This migration could be part of the normal physiologic migratory pattern of MSCs and/or be promoted by expression of the oncogenic EWS-FLI1 fusion.

Despite the absence of full transformation, EWS-FLI1+ stylopod cells reproducibly upregulated expression of pro-tumorigenic transcripts, *Dkk1*, *Prkcb*, and *Gli1*. These genes were not induced in the zeugopod cells (Summarized in Figure 6). These data suggest that regional differences in susceptibility to EWS-FLI1-induced transformation are likely to be dependent on the ability of the fusion to activate its downstream oncogenic program in the target cell. EWS-FLI1 induces transformation by reprogramming the epigenome, in particular through recruitment of epigenetic regulators that open chromatin and activate gene expression [28,30]. Significantly, early data suggests that despite its activity as a pioneer factor, EWS-FLI1 is unable to activate gene expression from areas of dense heterochromatin [33]. Therefore, a difference in chromatin state around key oncogenic genes could impact the ability of EWS-FLI1 to induce transformation in different cell populations. Furthermore, EWS-FLI1 is unable to induce transformation alone. A number of transcription factors, BMI-1 and Forkhead Box Q1 (FOXQ1) for example [34,35], have been shown to be required for EWS-FLI1 induced transformation. In addition, a new murine model of Shh-induced sarcoma revealed that activation of Gli1 in discrete populations of mesenchymal progenitors generated soft tissue tumors that displayed phenotypic and genotypic properties of human Ewing sarcoma, including high level expression of *Hoxd13*, in the absence of EWS-FLI1 [36]. This raises the possibility that differential activation of Gli1 in stylopod eSZ cells may be important for initiation of the transformation program. It remains to be determined whether regional differences in activation of Gli1 or other oncogenic genes between stylopod and zeugopod eSZ cells are due to regional differences in Hox expression.

In summary, MSCs from anatomically distinct sites are differentially susceptible to EWS-FLI1-induced transformation (Figure 6). These findings support the premise that the dominant presentation of Ewing sarcoma in pelvic and stylopod bones is attributable to anatomically-defined differences in MSCs, differences that are likely to be defined by embedded developmental transcription programs.

## 4. Materials and Methods

### 4.1. Staining of E18.5 eSZ

Forelimbs of E18.5 *Hoxa11^GFP/+^*; *Pthrp-lacz* embryos were formalin fixed and embedded in OCT. 18 micron sections were collected on slides. GFP was imaged followed by beta-galactosidase staining to visualize *lacz* expression. Antibody staining was separately completed by blocking tissue section with 10% normal donkey serum for one hour, and then incubating with rabbit anti-lacz antibody overnight. Tissue was washed and incubated with donkey anti-rabbit cy3 for 1 h. Nuclei were counterstained with dapi.

### 4.2. Isolation and Transduction of eSZ

Modified from Tanaka, JCI 2014, cells were isolated from both the stylopod and zeugopod of E18.5 embryos of CD1 and CD1/Bl6 litters. Cells were transduced with pCLS-EGFP empty or pCLS-EWS-FLI1-V5-2A-EGFP lentivirus. Lentivirus was produced in the University of Michigan Vector Core (Ann Arbor, MI, USA). IMDM 15% FBS media was supplemented with polybrene and Z-VAD. Cells were plated at 7.5 × 10^5^ cells per well in a 6 well plate and spun at 800× *g* for 20 min at 37 °C followed by 20 min at 37 °C 5% CO_2_ twice. Transduction was repeated the following day with fresh virus and media.

*Hoxd13^neo/neo^* cells were obtained from crossing *Hoxd13^neo/WT^* mice to *Hoxd13^neo/WT^* mice to generate a litter of ¼ *Hoxd13^WT/WT^*, ¼ *Hoxd13^neo/WT^*, and ½ *Hoxd13^neo/neo^*. eSZ isolations were completed as described above. The generation of *Hoxd13* mutant mice has been previously reported [37]. All procedures were by a protocol approved by the Institutional Animal Care and Use Committee and animal care was overseen by the Unit for Laboratory Animal Medicine (IACUC, #PRO00008750, renewal approved 20 November 2018).

Cell counting was done manually with trypan blue (1:1 dilution) and a hemocytometer.

### 4.3. RNA Isolation

RNA was collected from pelleted cells using the Zymo kit for RNA collection. cDNA was generated using the iSCRIPT cDNA synthesis kit (BioRad, Hercules, CA, USA).

### 4.4. qPCR

The following Taqman assays from Thermo Fisher (Waltham, MA, USA) were utilized in real-time quantitative PCR: mouse Hprt Mm03024075_m1, mouse Hoxd10 Mm00442839_m1, mouse Hoxd11 Mm02602515_mH, mouse Hoxd13 Mm00433973_m1, mouse Dkk2 Mm01322146_m1, mouse Prckb Mm00435749_m1, mouse Gli1 Mm00494654_m1, and human EWSR1-F Hs03024497.

The following primer sequences were used with iTaq Universal SYBR Green Supermix (BioRad, Hercules, CA, USA): mouse Hprt (AGTCCCAGCGTCGTGATTAG, TTTCCAAATCCTCGGCATAATGA), mouse Pthrp (CAGCAGTGGAGTGTCCTGGTATTC, GTGGTTTTTGGTGTTGGGAGC) [15]. All quantitative PCR were run on a Roche Lightcycler 480 II in a 384 well plate. The housekeeping gene hypoxanthine guanine phosphoribosyltransferase (HPRT) was used as a control for all qPCR experiments. mRNA levels of HPRT were consistent across MSC isolations and remained unchanged after EWS-FLI1 transduction. Data are normalized to HPRT in order to demonstrate the relative level of expression of transcripts of interest compared to this reproducibly, highly expressed gene.

### 4.5. Flow Cytometry

Cells were suspended in PBS and 1% FBS in a 5 mL filter top tube. Accuri C6 with what C Flow Plus. Collected 10,000 events. Compared to a GFP negative control.

### 4.6. Imaging

Mirocgraphs of GFP+ eSZ cells were obtained on living cells using an inverted Olympus IX83 (Shinjuku, Tokyo, Japan). 20× images were collected 10 dpt and 21 dpt to monitor GFP expression as well as cell morphology. CellSens Dimension software (Ver. 1.14, Shinjuku, Tokyo, Japan) captured the images.

### 4.7. Statistics

Statistical analysis consisted of two tailed students *t*-tests or multiple comparison ANOVA when applicable. Legends indicate which tests were completed.

## 5. Conclusions

These data elucidate fundamental knowledge about Ewing sarcoma, both in respect to the identity of the cell of origin as well as the molecular pathways that underlie EWS-FLI1-induced transformation. Detailed investigation into in vitro cell transformation revealed that in fact progenitors isolated from distinct skeletal regions, yet the same embryonic time point, retained differential sensitivity to EWS-FLI1-induced transformation. We were unable to generate tumors with subcutaneous injection of the transduced eSZ cells into nude mice, and therefore characteristics of in vivo tumor growth were not included in our phenotype analysis. These data begin to explain the predominant clinical presentation of Ewing sarcoma in the pelvis and femur. Additional studies will elucidate the permissive nature of the Ewing sarcoma cells of origin.

Furthermore, our studies utilizing a *Hoxd13* null mouse model have facilitated a more complete understanding of the molecular mechanisms of murine Ewing sarcoma. Genetic loss of *Hoxd13* did not result in a significant change in cell viability or target oncogene expression. This suggests that *Hoxd13*, while critical for human Ewing sarcoma tumorigenicity, is not required in mouse EWS-FLI1 induced transformation. As described above, further studies will illuminate whether this is a feature of EWS-FLI1 induced transformation or simply a limitation of the mouse model.

## Figures and Tables

**Figure 1 cancers-11-00313-f001:**
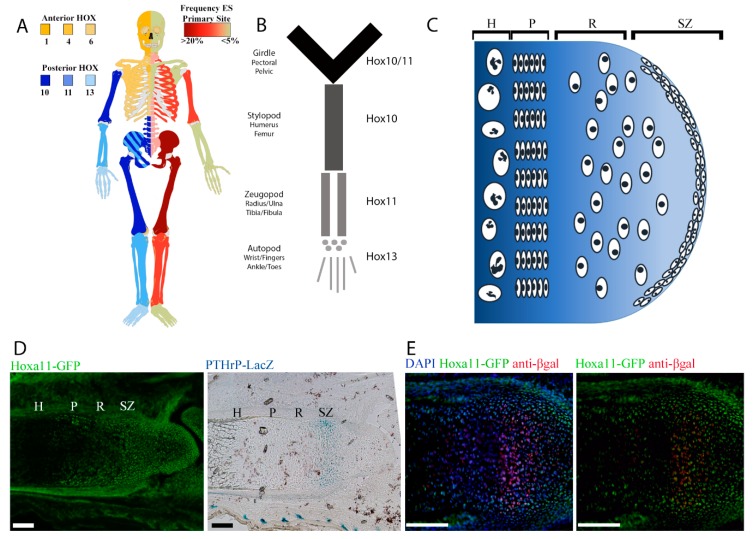
Human Ewing sarcoma localizes to skeletal compartments where posterior *HOX* genes are expressed. (**A**) Human skeleton demonstrating anterior (yellow) and posterior (blue) expression of *HOX* genes on the left contrasted with frequency of Ewing sarcoma primary site (red) on the right. (**B**) Anatomic terms for skeletal regions of the limbs. (**C**) Cartoon demonstrating the location of the embryonic superficial zone (eSZ) on the ends of the bones that undergo endochondral ossification. The eSZ lies above the growth plate: resting zone (R), proliferative zone (P), hypertrophic zone (H). (**D**) Micrographs of E18.5 zeugopod, Hoxa11-EGFP (green fluorescent protein) in green, Pthrp-Lacz in blue in an adjacent section. Scale bars represent 100 µm. (**E**) Micrographs of E18.5 zeugopod, Hoxa11-GFP in green, anti-βgal in red. Scale bars represent 100 µm.

**Figure 2 cancers-11-00313-f002:**
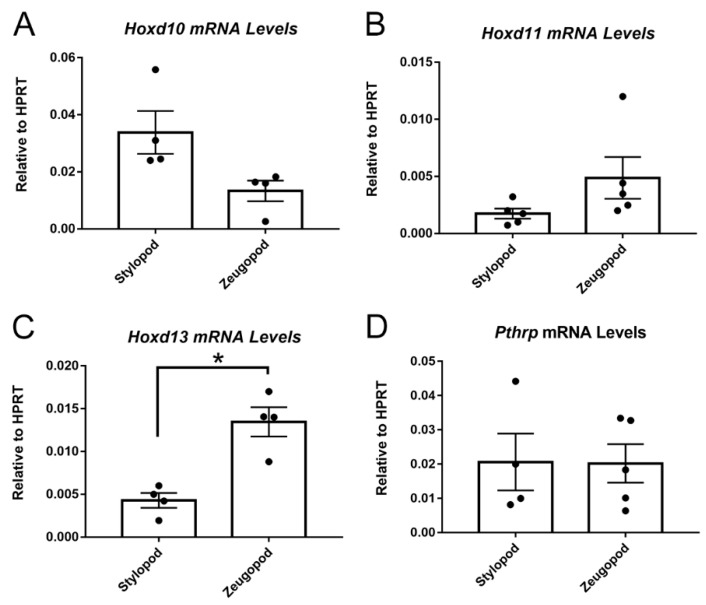
E18.5 stylopod and zeugopod eSZ cells express distinct Hox profiles. (**A**) Hoxd10 mRNA levels relative to hypoxanthine guanine phosphoribosyltransferase (HPRT) in freshly isolated stylopod and zeugopod eSZs. (**B**) Hoxd11 mRNA levels relative to HPRT in freshly isolated stylopod and zeugopod eSZs. (**C**) Hoxd13 mRNA levels relative to HPRT in freshly isolated stylopod and zeugopod eSZs. (**D**) Pthrp mRNA levels relative to HPRT in freshly isolated stylopod and zeugopod eSZs. * *p* ≤ 0.05. Error bars are SEM.

**Figure 3 cancers-11-00313-f003:**
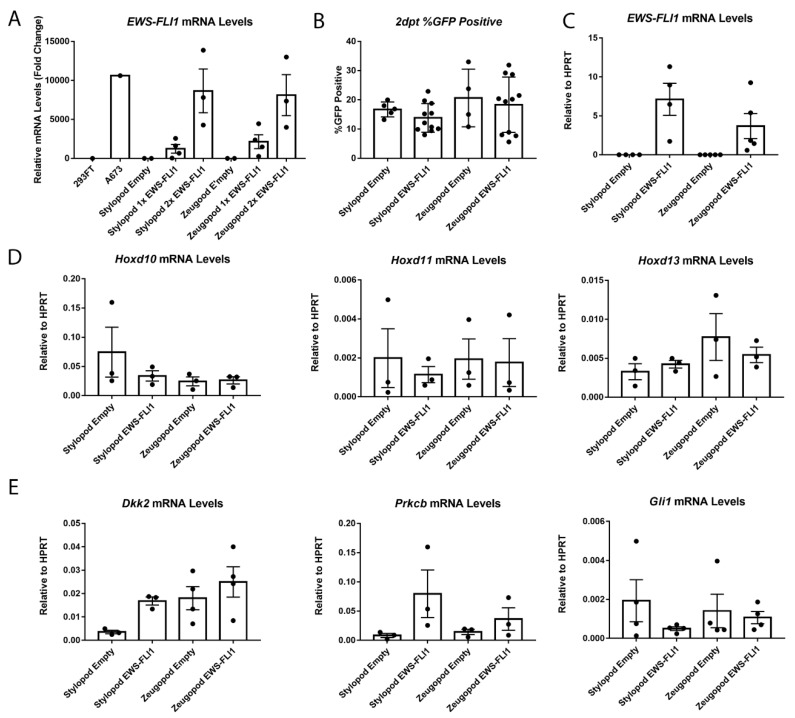
Stylopod and zeugopod eSZs undergo equivalent EWS-FLI1 transduction at two days post transduction (2 dpt). (**A**) Relative levels of *EWS-FLI1* expression with increasing doses of EWS-FLI1 lentivirus. A673 is human Ewing sarcoma cell line for comparison. (**B**) Flow cytometry for GFP in transduced eSZs. (**C**) *EWS-FLI1* mRNA levels relative to *HPRT*. (**D**) *Hoxd10*, *Hoxd11*, and *Hoxd13* mRNA levels relative to *HPRT*. (**E**) *Dkk2*, *Prkcb*, and *Gli1* mRNA levels relative to *HPRT*. Error bars are SEM.

**Figure 4 cancers-11-00313-f004:**
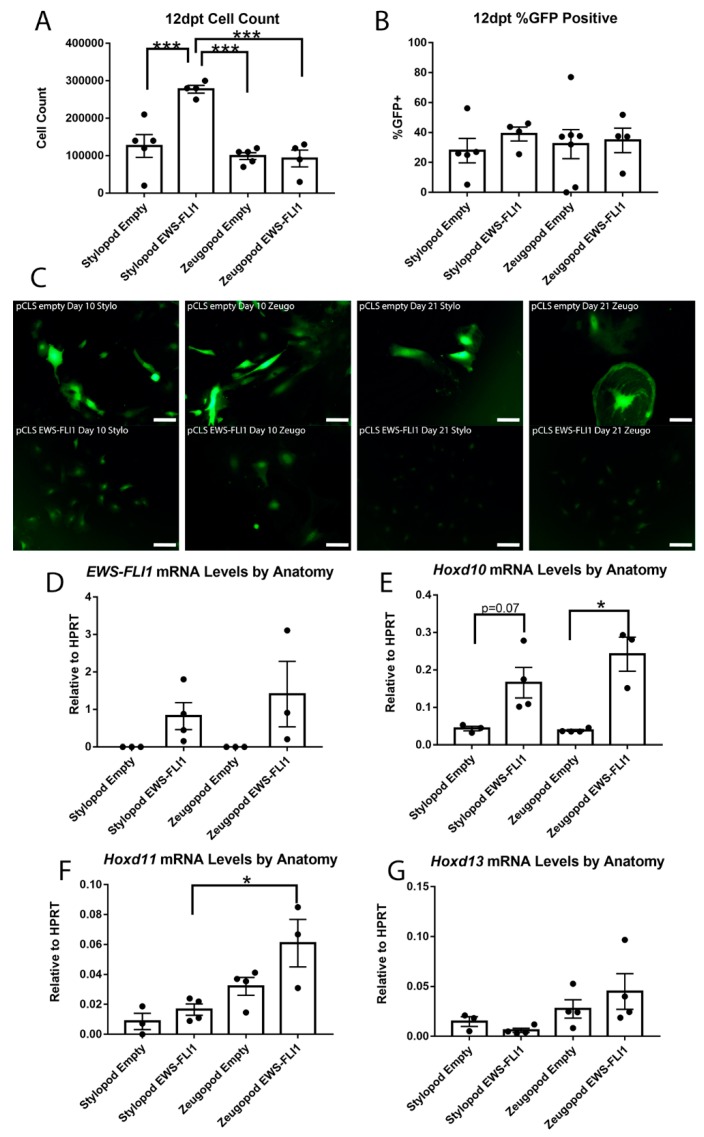
EWS-FLI1 changes cell phenotype in stylopod and zeugopod eSZs twelve days post transduction (dpt). (**A**) Cell count at 12 dpt. (**B**) Flow cytometry for GFP expression at 12 dpt. (**C**) Micrographs of GFP expressing cells with empty and EWS-FLI1 transduction at 10 and 21 dpt. Scale bars represent 50 µm. (**D**) *EWS-FLI1* mRNA levels relative to *HPRT*. (**E**) *Hoxd10* mRNA levels relative to *HPRT*. (**F**) *Hoxd11* mRNA levels relative to *HPRT*. (**G**) *Hoxd13* mRNA levels relative to *HPRT*. *** *p* ≤ 0.001; * *p* ≤ 0.05. Error bars are SEM.

**Figure 5 cancers-11-00313-f005:**
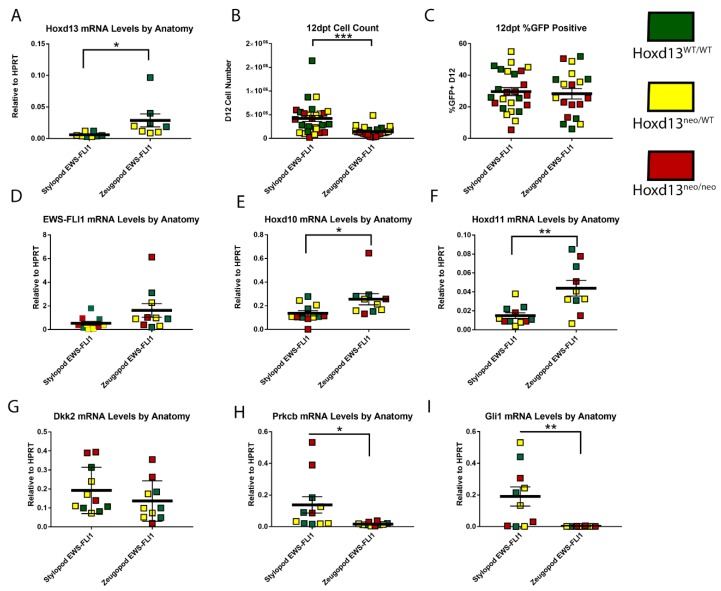
Loss of Hoxd13 does not impact EWS-FLI1 induced transformation by twelve days post transduction (dpt). (**A**) Hoxd13 mRNA levels relative to HPRT. (**B**) Cell count at 12 dpt. (**C**) Flow cytometry for GFP expression at 12 pt. (**D**) EWS-FLI1 mRNA levels relative to HPRT. (**E**) Hoxd10 mRNA levels relative to HPRT. (**F**) Hoxd11 mRNA levels relative to HPRT. (**G**) Dkk2 mRNA levels relative to HPRT. (**H**) Prkcb mRNA levels relative to HPRT. (**I**) Gli1 mRNA levels relative to HPRT. Green *Hoxd13^WT/WT^*, Yellow *Hoxd13^neo/WT^*, and Red *Hoxd13^neo/neo^*. *** *p* ≤ 0.001; ** *p* ≤ 0.01; * *p* ≤ 0.05. Error bars are SEM.

**Figure 6 cancers-11-00313-f006:**
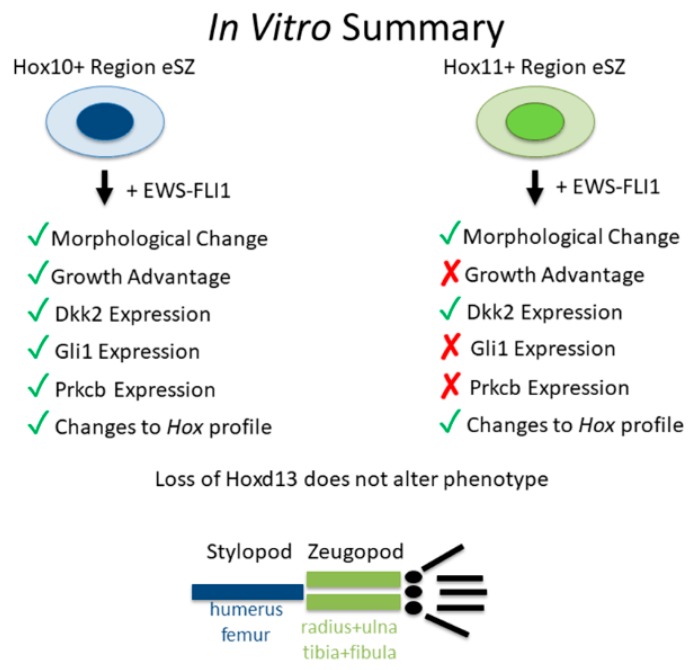
Summary of in vitro data.

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
