# Peer review of "Anatomic Origin of Osteochondrogenic Progenitors Impacts Sensitivity to EWS-FLI1-Induced Transformation"

_cancers, 2019, doi:10.3390/cancers11030313_

Round 1

Reviewer 1 Report

The article entitled “Anatomic Origin of Osteochondrogenic Progenitors Impacts Sensitivity to EWS-FLI1-induced Transformation” by Pfaltzgraff ER, et al. is a study focusing on an anatomical origin of the progenitor cell of Ewing sarcoma. The authors hypothesized that Hox genes in posterior skeleton mesenchymal stem cells (MSCs) contributed to Ewing sarcoma tumorigenesis because of the propensity for Ewing sarcoma to arise in the posterior skeleton. Their hypothesis is interesting and seems reasonable. This paper is well-organized and will be of interest to readers of Cancers. I have only a few concerns mentioned below.

Comments

1. It is reasonable to interpret that the results of this study indicate that the cell of origin of Ewing sarcoma is a MSC located in the embryonic superficial zone (eSZ) of the stylopod bones. As the authors say, Ewing sarcoma occurs predominantly in the stylopod bones, however, the favorite site is the diaphysis of the stylopod bones, not the epiphysis. Could you explain this discrepancy about the exact location of occurrence? Some explanations should be added in Discussion.

2. As the authors mention, Hox genes expression would be associated with the initiation of Ewing sarcoma because the difference between eSZ cells in the stylopod and zeugopod bones includes HOXD10 and HOXD11 expression. To confirm those results, knockout or knockdown of Hox genes, not only HOXD13 but also HOXD10, would be of great importance. The authors are encouraged to add experiments to see the influence of HOXD10 silencing. 

Author Response

·         Comment 1: It is reasonable to interpret that the results of this study indicate that the cell of origin of Ewing sarcoma is a MSC located in the embryonic superficial zone (eSZ) of the stylopod bones. As the authors say, Ewing sarcoma occurs predominantly in the stylopod bones, however, the favorite site is the diaphysis of the stylopod bones, not the epiphysis. Could you explain this discrepancy about the exact location of occurrence? Some explanations should be added in Discussion.

Response: (added to Discussion, Lines 305-316) The anatomic presentation of tumors in the diaphysis of human stylopod bones is intriguing, given the marked enrichment of the Hox+ MSC in the epiphysis. We hypothesize that there are two potential explanations for this discrepancy. First, although enriched in the embryonic superficial zone, Hox+ MSC are also present along the perichondrium of the diaphysis (see Figure 1D and Ref. 21), and these cells would also be predicted to be susceptible to transformation. It is conceivable that the microenvironment of perichondrial MSCs is more conducive to cell transformation than that of epiphyseal MSCs, especially given proximity of diaphyseal MSCs to bone marrow, a tumor niche that favors Ewing sarcoma growth in human patients. A second possibility is that the cells of origin reside in the epiphyseal region at the time of the initial translocation event but that they migrate to the diaphysis. This migration could be part of the normal physiologic migratory pattern of MSCs and/or be promoted by expression of the oncogenic EWS-FLI1 fusion.

·         Comment 2: As the authors mention, Hox genes expression would be associated with the initiation of Ewing sarcoma because the difference between eSZ cells in the stylopod and zeugopod bones includes HOXD10 and HOXD11 expression. To confirm those results, knockout or knockdown of Hox genes, not only HOXD13 but also HOXD10, would be of great importance.  The authors are encouraged to add experiments to see the influence of HOXD10 silencing

Response: We completely agree with the reviewer that the next logical step is to interrogate the functional contribution of stylopod-specific Hox genes, in particular Hoxd10, and we are keen to pursue this question. Indeed, in the context of human Ewing sarcoma, knockdown of HOXD10, HOXD11, and HOXD13 all independently impaired the growth and invasive properties of tumor cells, demonstrating that all members of the posterior HOXD cluster contribute to the tumorigenic phenotype (Ref. 18). Whether they work in concert or independently remains to be determined. Given our findings that stylopod cells are most susceptible to EWS-FLI1-induced transformation, we are very interested to next study Hoxd10. However, to address the requirement for Hoxd10 in the context of murine stylopod-derived MSCs, Hoxd10 knockout mice will be required.

We respectfully submit that these studies are beyond the scope of the current manuscript, for which only minor revisions have been requested. To directly respond to the question in the revised manuscript, we have added the comments above to the discussion (see Lines 295-304).

Reviewer 2 Report

This is a well-written, informative manuscript which details how the anatomic origin of osteochondrogenic progenitors potentially impacts sensitivity to EWS-FLI1-induced transformation and therefore explains the anatomic distribution of Ewing sarcoma, which most commonly arises in the posterior skeleton. The authors hypothesize that expression of posterior HOXD genes may increase cellular tolerance for EWS-FLI1 induced transformation, which is cytotoxic in other anatomic compartments with different gene expression. Their data show that stylopod-derived eSZ cells show morphological changes and avoidance of senescence in response to EWS-FLI1 transduction. These cells also showed upregulation of EWS-FLI1 target gene expression. Despite these supporting data elements, the cells did not demonstrate transformation when assayed by injection into nude mice.

Although the cellular population of interest was not completely transformed by EWS-FLI1 transduction, the data presented are extremely important to the field and informative for the scientific community. In addition, demonstration that loss of Hoxd13 (a posterior Hox gene) did not impact susceptibility to EWS-FLI1-induced transformation is a very important negative data point when considering the current hypothesis.

I recommend that this manuscript be published with the following minor revisions:

1.       Gene expression data are normalized to HPRT, but this is never explained in the manuscript. I understand that this is a convention of the field and the study of bone/development. However, I think it would be prudent to define HPRT and why it is used to normalize data in a single sentence somewhere in the methods section of the manuscript.

2.       Figure 1A is an extremely helpful orienting graphic. However, the yellow and blue scales on the anterior and posterior HOX expression legends seem to be inverted? Please double check this an ensure that it is appropriate. For example, Ewing sarcoma has a propensity for the femur, which has increased posterior HOX gene expression. However, if you look at the figure, the dark blue color associated with the femur shows decreased expression (10 versus 13) in the femur?

3.       Because the main point of Figure 4C is that EWS-FLI1 transduction results in a change in cellular morphology in stylopod versus zeugopod derived eSZ cells, scale bars should be utilized on this figure. In addition, Figure 1D and E would benefit from inclusion of scale bars.   

Author Response

We thank the reviewers for their thorough reading and critical appraisal of our manuscript. We agree completely with the comments and have made modifications to the revised manuscript to address all concerns. Specifically, the following revisions have been made in response to reviewer 2: 

·         Comment 1: Gene expression data are normalized to HPRT, but this is never explained in the manuscript. I understand that this is a convention of the field and the study of bone/development. However, I think it would be prudent to define HPRT and why it is used to normalize data in a single sentence somewhere in the methods section of the manuscript.

Response: We thank the reviewer for this suggestion to clarify our rationale. In response, we have added the following statement to the methods (Line 376-380): The housekeeping gene hypoxanthine guanine phosphoribosyltransferase (HPRT) was used as a control for all qPCR experiments. mRNA levels of HPRT were consistent across MSC isolations and remained unchanged after EWS-FLI1 transduction. Data are normalized to HPRT in order to demonstrate the relative level of expression of transcripts of interest compared to this reproducibly, highly expressed gene.

·         Comment 2: Figure 1A is an extremely helpful orienting graphic. However, the yellow and blue scales on the anterior and posterior HOX expression legends seem to be inverted? Please double check this an ensure that it is appropriate. For example, Ewing sarcoma has a propensity for the femur, which has increased posterior HOX gene expression. However, if you look at the figure, the dark blue color associated with the femur shows decreased expression (10 versus 13) in the femur?

Response:  We thank the reviewer for asking us to improve the graphic to better represent the intended message. As s/he points out, the original gradient was meant to show expression of specific HOX genes, but instead it appeared as a gradient of high to low expression. This was not our intention. In response, we have changed the key to Figure 1A to indicate that different colors represent different anatomically defined HOX genes rather than a gradient of expression.

·         Comment 3: Because the main point of Figure 4C is that EWS-FLI1 transduction results in a change in cellular morphology in stylopod versus zeugopod derived eSZ cells, scale bars should be utilized on this figure. In addition, Figure 1D and E would benefit from inclusion of scale bars.

Response: We regret the oversight on the original submission. The panels have been amended to include scale bars.